# Food Beliefs and the Risk of Orthorexia in Patients with Inflammatory Bowel Disease

**DOI:** 10.3390/nu16081193

**Published:** 2024-04-17

**Authors:** Francesca Maria Di Giorgio, Stefania Pia Modica, Marica Saladino, Stefano Muscarella, Stefania Ciminnisi, Piero Luigi Almasio, Salvatore Petta, Maria Cappello

**Affiliations:** 1Gastroenterology and Hepatology Section, Department of Health Promotion, Mother and Child Care, Internal Medicine and Medical Specialties (PROMISE), University of Palermo, Piazza delle Cliniche, 2-90127 Palermo, Italystefimodica97@gmail.com (S.P.M.); muscarellastefano@gmail.com (S.M.);; 2Department of Health Promotion, Mother and Child Care, Internal Medicine and Medical Specialties (PROMISE), University of Palermo, Piazza delle Cliniche, 2-90127 Palermo, Italy

**Keywords:** inflammatory bowel disease, orthorexia nervosa, eating disorders

## Abstract

Patients with inflammatory bowel disease (IBD) believe that diet plays a significant role in the pathogenesis of their disease and the exacerbation of their symptoms. They often adopt restrictive diets that can lead to malnutrition, anxiety, and stress. Recent studies have found a correlation between IBD and eating disorders, such as anorexia nervosa and ARFID (Avoidant Restrictive Food Intake Disorder). None of these studies report an association with orthorexia nervosa, which is an obsession with healthy and natural foods. The aim of this study was to assess the risk of orthorexia nervosa in patients with IBD. A total of 158 consecutive subjects were recruited, including 113 patients with IBD and 45 controls. The standardized Donini questionnaire ORTO-15 was administered to assess the risk of orthorexia, and clinical and demographic data were collected. The results showed that patients with IBD had a risk of developing orthorexia nervosa of 77%. This was significantly higher than the 47% observed in the control group. In the patients with IBD, the risk of orthorexia was associated with a lower BMI, at least in patients older than 30 years, and it was also associated with marital status in patients younger than 30. In conclusion, many patients with IBD are at increased risk of developing orthorexia nervosa, which may have a negative impact on their psychological wellbeing and social sphere, expose them to a high risk of nutritional deficiencies, and affect their overall quality of life. Further high-quality studies are needed to assess the clinical impact of orthorexia and its correlation with clinical features and classified eating disorders.

## 1. Introduction

Inflammatory bowel disease (IBD) is a chronic disease that manifests in two main forms: Crohn’s disease (CD), which can affect any part of the digestive tract, and ulcerative colitis (UC), which affects only the colon [1,2]. The etiopathogenesis of these diseases is not yet known; although, several factors are thought to be involved, including predisposing genetic factors, environmental factors, the role of the intestinal microbiota, the function of the intestinal epithelial barrier, and the innate and specific immune response [1,2]. Among the environmental factors, diet might play a relevant role. Furthermore, patients with IBD often have the perception that food can worsen their disease, and their excessive focus on the quality, quantity, and type of food that they intake can lead to unhealthy habits and, in some cases, caloric–protein malnutrition [3,4].

The medical literature has recently focused on the role of diet in IBD, a topic that has been neglected in the past. The International Organization for the Study of Inflammatory Bowel Diseases (IOIBD) consensus has formulated dietary guidance regarding specific dietary components, food groups, and food additives that it may be prudent to increase or decrease in the diet of patients with IBD to control and prevent relapse: the authors recognize that the available evidence is of varying quality and some derives from animal studies; however, they advise reducing saturated and trans fats, the intake of red/processed meats and myristic acid (palm oil, coconut oil, dairy fats), and the exposure to maltodextrins, carrageenans, carboxymethylcellulose, polysorbate-80, titanium dioxide, and other nanoparticles [5]. Recent studies demonstrate symptomatic benefits in patients with IBD who are on reduced carbohydrate diets [6] or diets based on the Mediterranean model [7]. Considering all of this evidence, the ESPEN (European Society for Clinical Nutrition and Metabolism) guidelines [8] provide extensive recommendations in the field of nutrition for IBD.

To prevent IBD, the ESPEN recommends the following:a diet rich in fruits and vegetables, enriched in omega-3 fatty acids, and low in omega-6;the exclusion of ultra-processed food and dietary emulsifiers;breastfeeding, as breast milk is the optimal food for infants and reduces their risk of developing IBD.

On the other hand, to prevent malnutrition and micronutrient deficiencies in individuals with IBD, the ESPEN recommends the following:nutritional screening at the time of diagnosis and on a regular basis thereafter;adequate treatment of malnutrition when documented, as it is associated with a worse prognosis, an increased rate of complications, mortality, and a poorer quality of life;iron supplementation in all patients with IBD in the presence of deficiency anemia.

As far as the role of diet in the treatment of IBD is concerned, on the basis of the available evidence, the ESPEN states that there is no “IBD diet” that can promote or maintain remission in patients with IBD, but recommends that patients with IBD in remission be supervised by a dietitian to improve their nutritional therapy [8].

Previous studies have shown that individuals with IBD are at increased risk of developing restrictive eating patterns and eating disorders, such as anorexia nervosa and ARFID [9]. Our group recently conducted a review on eating disorders in IBD [10]. In contrast, there are no studies dedicated to the correlation between IBD and orthorexia nervosa.

The term orthorexia nervosa derives from the Greek words “ortho”, meaning “correct” or “right”, and “orexis”, meaning “appetite” or “desire to eat”. Dr. Steven Bratman first used this term in 1997 [11]. It describes a condition characterized by an eating behavior that follows a pathological obsession with biologically pure and healthy nourishment, leading to a search for natural foods, free from artificial chemicals, and an excessive scrutiny of food origin, processing, and packaging [11]. The most avoided types of food include those containing preservatives, additives, colorings, excess fat, excess sugar, and excess salt. [12,13]. These behaviors cause the avoidance of many types of nutrients, resulting in nutritional deficiencies [14]. This may lead to a dietary depletion and a shift away from a dietary pattern based on balance, variety, and nutritional compensation. Some of the most commonly observed effects include caloric malnutrition, vitamin deficiencies, osteoporosis, and muscle atrophy. The prevalence rate of orthorexia is on average 6.9% for the general population and 35–57.8% for high-risk groups such as dietitians, healthcare-related professionals including medical students, people who participate in sports, and performance artists [14]. The variability of this condition depends on various factors, including diagnostic criteria and cut-offs, as well as cultural, geographic, and gender differences [15]. Individuals with orthorexia spend a significant amount of time thinking about food, and planning, purchasing, preparing, and consuming meals containing nutrients that they consider healthy. Any deviation from their self-imposed rules can induce feelings of shame, guilt, and anxiety, and lead to dietary restriction [11,12,14,15].

Orthorexia is a relatively new phenomenon and to date it is still not formally recognized as an eating disorder and is not included in the official ICD-11 and DSM-V classifications, and neither the American Psychiatric Association nor the World Health Organization officially recognize orthorexia nervosa as a mental disorder. However, some consider it to be a subtype of ARFID. Other researchers believe it should be treated as a disorder concerning abnormal eating behavior linked with obsessive compulsive disorder [12]. To date, there is no gold standard for diagnosing orthorexia nervosa, although several tools have been proposed. One of the most widely used diagnostic tools is the ORTO-15 test, designed in 2005 by Donini et al. [16,17].

## 2. Material and Methods

From February to October 2022, 158 consecutive subjects, including 113 patients diagnosed with IBD and 45 controls, were recruited prospectively at the Gastroenterology and Hepatology Unit of the University Hospital of Palermo. Specifically, the cohort of patients with IBD consisted of 74 patients receiving biologic drug treatments (Infliximab, Vedolizumab, Adalimumab, Ustekinumab), 28 patients receiving conventional therapy, and 12 patients admitted to the Gastroenterology Unit for disease flares. All patients met the following inclusion criteria: aged over 18 years, with an established diagnosis of UC or CD, and consented to voluntary participation in the study. The following data were collected: clinical and demographic data (age, sex, family history of IBD, smoking habits, diagnosed food intolerances), marital status, educational qualifications and occupational activity, Body Mass Index (BMI), the average amount of water consumed per day, and disease characteristics (type of IBD, activity status, any previous surgeries, and the presence of an ostomy), including concomitant therapy. The disease activity status at the time of enrollment was assessed using the partial Mayo score for UC (Figure 1) and the Harvey Bradshaw Index (HBI) for CD (Figure 2). The control group (45 healthy subjects) consisted of 25 individuals including trainees and employees at the Azienda Ospedaliera Universitaria “Paolo Giaccone” of Palermo and 20 individuals external to the medical institution, recruited among relatives of trainees who agreed to fill out the questionnaire. Both the IBD patients and controls agreed to participate in the study. The study involved the administration of the standardized ORTO-15 test questionnaire, validated by Donini et al. [16,17] (Figure 3). The questionnaires were administered by two qualified dietitians, S.C. and S.P.M., who clarified each question to the patients and collected the answers in a shared Excel file.

This study has been approved by the Ethics Committee—Palermo 1 (Approval Report no. 2/15 February 2022).

The ORTO-15 scale provides an assessment of behaviors related to food choice and the purchase, preparation, and consumption of food.

The questionnaire consists of 15 items. Questions are separated based on eating behaviors: cognitive (items 1, 5, 6, 11, 12, and 14), clinical (items 3, 7, 8, 9, and 15), and emotional (items 2, 4, 10, and 13). For each question, the patient has 4 response options (always, often, sometimes, never), and to each response a score is given ranging from a minimum of 1 point to a maximum of 4 points. The cut-off validated by Donini for the diagnosis of orthorexia nervosa is a score < 40.

Statistical analyses were performed using IBM SPSS software (Version 29.0.2.0 (20)). Continuous variables were expressed as the mean ± standard deviation, and categorical variables were expressed as frequencies and percentages. The characteristics of the patients with IBD were compared with those of the controls, and within the group of patients with IBD, the characteristics of those with orthorexia were compared with those of their counterparts. Student’s *t*-test was used to compare continuous variables, and the chi-square test was used to compare categorical variables. A difference with a *p*-value < 0.05 was considered statistically significant.

## 3. Results

Out of the 113 patients who participated in the study, 48 (42%) had UC and 65 (58%) had CD. The mean age was 50 ± 16 years, ranging from 18 to 84 years, with a slight prevalence of the male sex (54%) and a mean BMI of 25 ± 5 kg/m^2^. Most patients were Italian (98%) and married (65%); 42% had a high-school diploma, while only 14% had a university degree. At the time of interviewing, 46% of patients were employed, 31% were unemployed, 18% were retired, and 8% were students. The diagnosis of IBD was made within the 5 years prior to the interview in about 40% of patients and between 5 and 20 years prior to the interview in 49.5%, while only 11.5% of patients had been diagnosed with the disease for more than 20 years. In terms of disease activity, as assessed by the partial Mayo Score and HBI, 58% were in remission, 27% had mild disease activity, 13% had moderate activity, and only 2% had severe activity. Moreover, 16% of the patients had previously undergone resection surgery for IBD and 2.6% had an ostomy. Regarding therapy, 75% of the patients were being treated with biologics, while 25% were on conventional therapy. In addition, 5.4% of the patients had a diagnosed food intolerance (lactose or gluten), 66% smoked, and 47% drank more than 2 liters of water per day.

The control group consisted of a sample of 45 subjects, 60% of whom were female, with a mean age of 32 ± 12 years ranging from 18 to 58 years, and a mean BMI of 23 ± 4 Kg/m^2^. This population was predominantly (69%) unmarried with only a minority (31%) being married. All of them had an educational level higher than primary school (3% had a secondary-school degree, 55% had a high-school degree, and 38% had a university degree). In total, 44% of these individuals had a regular job, while 16% were unemployed, none were retired, and 40% were still students. In this group, 9% had a diagnosed food intolerance (lactose/gluten), 64% smoked, and 44.5% drank more than 2 liters of water per day.

### 3.1. Comparison between the IBD Patients and Control Group

Comparing the IBD patient population with the control group demonstrates that the patients with IBD were significantly older, had a higher BMI, and were more likely to be married, have a lower educational level, and have a lower employment rate (Table 1).

On the basis of the results of the Donini questionnaire with the cut-off of 40, we found that 77% of patients with IBD had a tendency to develop orthorexia, significantly higher than the 47% observed in the control group (*p* < 0.001).

### 3.2. Comparison between IBD Patients with the Risk of Orthorexia and without the Risk of Orthorexia

Focusing the analysis on the group of patients with IBD, no statistically significant differences were found between the patients with and without the risk of orthorexia in terms of their age (*p* = 0.66), gender (*p* = 0.37), marital status (*p* = 0.78), educational level (*p* = 0.88), and occupation (*p* = 0.64), although a non-significant trend towards lower BMI levels was observed in the patients at risk of orthorexia compared to those without the orthorexia risk (mean BMI 26.7 vs. 25 Kg/m^2^, *p* = 0.12). Regarding IBD characteristics, the prevalence of orthorexia did not differ in relation to the diagnosis of UC or CD (*p* = 0.98), nor in relation to the disease duration (*p* = 0.75), the type of biological or conventional therapy (*p* = 0.42), the disease activity as measured by the Mayo score/HBI (*p* = 0.47), or the presence of an ostomy (*p* = 0.33). The only statistically significant difference observed was in the history of previous surgery for IBD: in fact, a 19.5% prevalence of previous surgery was observed in the patients at risk of orthorexia, which was significantly higher than the 3.8% prevalence observed in the patients not at risk of orthorexia (*p* = 0.05) (Table 2). However, after stratifying the IBD patients into two age groups (<30 years old and >30 years old), surgery was no longer found to be associated with the risk of orthorexia. It is worth noting that there was a trend towards a significant relationship between orthorexia and lower BMI levels in older patients, and a significant relationship between the tendency towards orthorexia and marital status in younger patients (Table 3 and Table 4). The mean score of the ORTO-15 test for the IBD patients at risk of orthorexia was 34.44, while in the IBD patients without the risk of orthorexia it was 41.

The percentual distribution of the responses to the ORTO-15 questionnaire in the patients and controls is reported in the Appendix A.

## 4. Discussion

In this study of a prospective cohort of patients with IBD, we have demonstrated that the risk of orthorexia among these patients is high, at 77%, and significantly higher than that observed in a control group.

To our knowledge, there are no studies that have assessed the prevalence of orthorexia nervosa in patients with IBD, and our study is the first to document an increased risk of orthorexia in these patients, with a significantly higher rate than in a control population. In the context of digestive diseases, similar results were found in a study conducted in women with celiac disease by Kujawowicz and coworkers [18]. Another study conducted on Hungarian university students addressed the risk of orthorexia in irritable bowel syndrome. Their study found that functional gastrointestinal symptoms were positively associated with symptoms of orthorexia nervosa and emotional eating. The relationship between functional gastrointestinal symptoms and symptoms of orthorexia nervosa was partially mediated by health anxiety [19]. In this setting, another paper highlighted a bidirectional relationship, since IBS symptoms are more common in patients with established eating disorders [20]. Such bidirectionality with anxiety and depression has already been described in IBD, at least for anorexia nervosa [21]. Anxiety and depression [22], whose prevalence has been estimated in IBD to be as high as 29–35% during remission and 60–80% during relapse, may be the driving force for IBD patients to cross the “fine line” between a healthy eating belief aimed at promoting physical and mental health and orthorexia, i.e., an obsessive and even harmful pursuit of “pure” food.

Our study also investigated whether demographic, clinical, and social factors, as well as factors related to chronic IBD, are associated with the risk of orthorexia. In this regard, we observed that IBD patients with the orthorexia risk tended to have a lower BMI, though this result was not statistically significant. Those who have an orthorexic tendency, in contrast to other eating disorders, are not scared of becoming fat; their goal is to consume a healthy diet, and losing weight is just a consequence [15]. Most studies have not found any association between orthorexia nervosa and BMI [15]. Indeed, when our patients were stratified into two age groups, this association could be confirmed only in patients > 30 years old. We have also documented that the prevalence of orthorexia is significantly higher among those who have undergone surgery for their IBD. This finding may depend on the fact that those who undergo surgery have a more severe disease course and a more challenging life experience, which leads them to modify their diet as well. This association was not observed when the patients were stratified into age groups. Further analyses of these data may be necessary in larger cohorts.

No difference was observed in relation to the type of IBD, the severity of disease and type of therapy, or sociocultural and occupational status. The final observation is somewhat unexpected since the preoccupation for healthy food would suggest more costly food choices. Indeed, high-risk groups for orthorexia have been reported to be university students, people who practice sports, doctors, and performance artists, while only 14% of our study population had a university qualification. An alternative interpretation is that patients with IBD have a strong belief in the role of food in managing their condition, regardless of their economic status.

We found no difference within the patients at risk of orthorexia and the patients not at risk as far as concerns their gender. Indeed, previous studies have reported discrepant results on the association between gender and a tendency towards orthorexia [15].

Also, marital status had no relationship with the risk of orthorexia in our patients, except for the younger patients, while other studies have reported a higher risk in people who live alone and single people [23]. However, a Turkish study identified being married as a risk factor [24].

Our study has identified a new form of disordered eating in a relatively large sample of IBD patients. Although not yet classified in DSM-V classifications, orthorexia could be part of the spectrum of eating disorders. It is warranted to identify the development of orthorexic behavior in IBD patients in a timely manner to prevent nutritional consequences and the impact on quality of life. Food is not just about nourishment; it is also a source of pleasure, cultural heritage, and social connection.

Our study has some limitations, particularly in relation to the cohort of patients with IBD who were followed in a tertiary referral center for IBD and mostly treated with biologic drugs. Therefore, this cohort is not representative of all patients with IBD. In addition, the small sample size of the control population may affect the observed results and their interpretation. This study could also have been limited by the possibility of biased responses due to inaccuracies in the way the questionnaires were administered, although the support of experienced dietitians for the administration of the questionnaires should have overcome any such bias.

Finally, caution should be exercised when interpreting the differences in the prevalence of orthorexic tendencies observed compared to the control population, depending on the differences in age, BMI, and sociocultural status between the patients with IBD and our control group. However, the results in the control population are in line with a previous Italian study carried out in students and university employees of the University of Pisa [25].

The choice of the cut-off for the diagnosis of orthorexia has been questioned in previous studies [18]. However, a validation study has shown that a cut-off of 40 is the most predictive (sensitivity: 100.0%, specificity: 73.6%, positive predictive value: 17.6%, negative predictive value: 100.0%) [19]. The ORTO test itself does not measure clinical impairments caused by an excessive preoccupation with healthy eating, such as extreme weight loss, social impairment, and occupational impairment, which may explain why the detected prevalence rates for orthorexia are so high. Other research tools for the diagnosis of orthorexia are under investigation [26,27].

Finally, a further limitation is that the use of only one questionnaire evaluating orthorexic features did not allow us to evaluate the relationship between orthorexia and other psychopathological disorders [22]. In conclusion, our study showed that patients with IBD may have an increased risk of developing orthorexia, with a 77% prevalence rate in our study group. In our study group, this risk was associated with a lower BMI and a history of previous IBD surgery. Orthorexia may negatively impact a patient’s psychological wellbeing and social sphere, increase their risk of developing severe nutritional deficiencies, and affect their overall quality of life. Given the high prevalence of orthorexia and other eating disorders, it is important to have an experienced dietitian and dedicated psychologist as part of the multidisciplinary team responsible for the patient’s care to provide appropriate nutritional and psychological counseling. Rebuilding a healthy relationship with food is crucial for overall wellbeing. Cognitive behavioral therapy or participating in support groups or group therapy sessions could be helpful therapeutic strategies.

## 5. Conclusions

In conclusion, our study shows that patients with IBD are at increased risk of orthorexia, an eating behaviour not yet classified as a classical eating disorder but associated with potentially harmfyl dietary restrictions. Further high-quality studies in larger cohorts are needed to confirm our results and evaluate the clinical impact of orthorexia in patients with IBD in order to identify the predictive factors for orthorexic behavior, as well as the tools required for its assessment and its management. Long-term follow-up evaluations of orthorexic IBD patients are necessary to investigate whether orthorexia develops into classified eating disorders.

## Figures and Tables

**Figure 1 nutrients-16-01193-f001:**
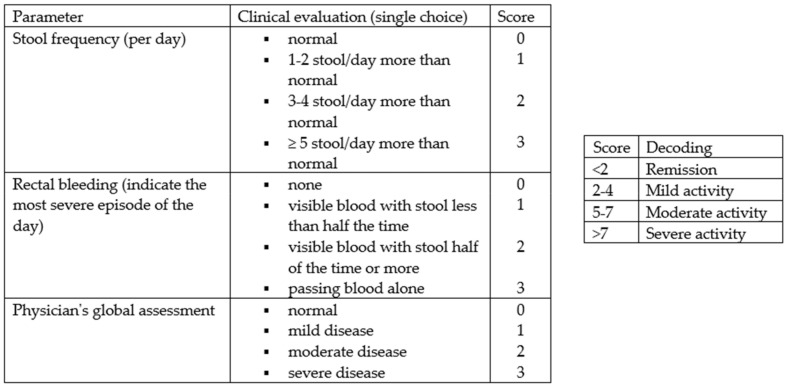
Partial Mayo score (igibdscores.it).

**Figure 2 nutrients-16-01193-f002:**
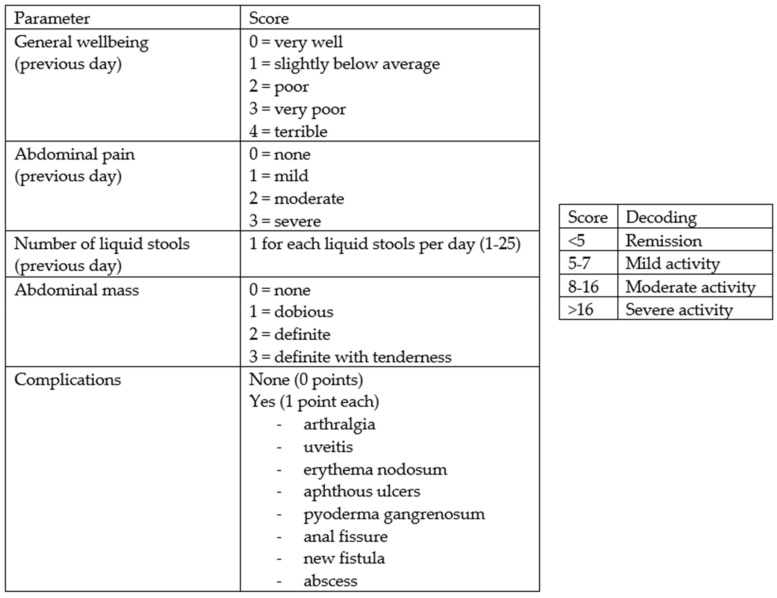
Harvey Bradshaw Index (HBI) (igibdscores.it).

**Figure 3 nutrients-16-01193-f003:**
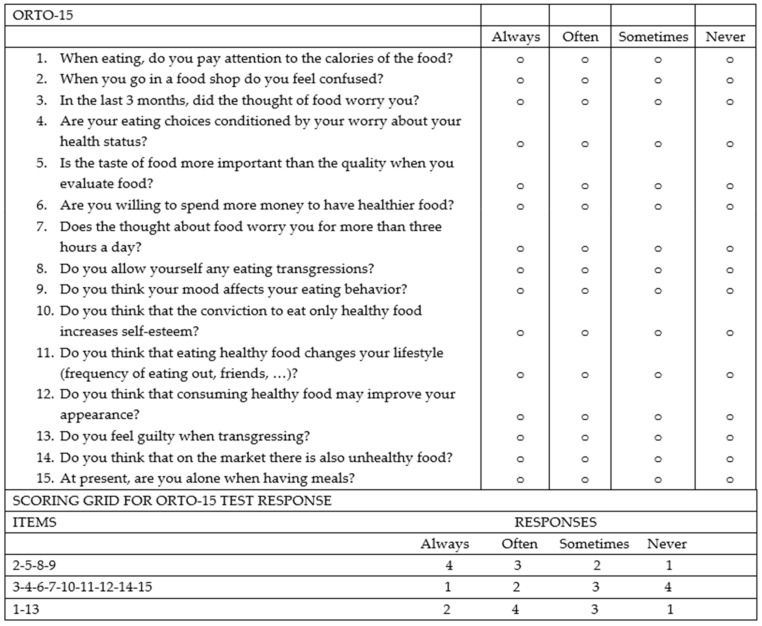
ORTO-15: the test for the diagnosis of orthorexia nervosa (Donini et al. 2005) [16].

**Table 1 nutrients-16-01193-t001:** Comparison between the IBD patients and control group at the time of recruitment.

	IBD Patients N = 113	Control Group N = 45	*p*-Value
Age (years) (mean ± SD)	50 ± 16	32 ± 12	<0.001
**Gender**			
Female	52 (46%)	27 (60%)	0.11
Male	61 (54%)	18 (40%)	
**Marital status**			
Married	73 (65%)	14 (31%)	0.003
Single/widowed/divorced	40 (35%)	31 (69%)	
BMI—Body Mass Index (mean ± SD)	25 ± 5	23 ± 4	0.004
**Educational qualifications**			
Primary-school degree	12 (11%)	0	<0.001
Secondary-school degree	37 (33%)	3 (7%)	
High-school degree	48 (42%)	25 (55%)	
University degree	16 (14%)	17 (38%)	
**Employment**			
Employed	52 (46%)	20 (44%)	<0.001
Unemployed	35 (31%)	7 (16%)	
Retired	18 (16%)	0	
Student	8 (7%)	18 (40%)	
**Diagnosis**			
CD	48 (42%)		
UC	65 (58%)		
**Disease duration**			
<5 years	44 (39%)		
5–10 years	26 (23%)		
11–20 years	30 (26.5%)		
>20 years	13 (11.5%)		
**Therapy**			
Biologic agents	85 (75%)		
Conventional therapy	28 (25%)		
**Partial Mayo score/HBI**			
Remission	65 (58%)		
Mild activity	31 (27%)		
Moderate activity	15 (13%)		
Severe activity	2 (2%)		
**Surgery**			
Yes	18 (16%)		
No	95 (84%)		
**Ostomy**			
No	110 (7.4%)		
Yes	3 (2.6%)		
**Food intolerances**			
No	107 (94.6%)	41 (91%)	
Lactose intolerance/celiac disease	6 (5.4%)	4 (9%)	0.45
**Smoking habit**			
Yes	75 (66%)	29 (64%)	
No	38 (34%)	16 (36%)	0.81
**Daily water consumption**			
<1 L	28 (25%)	10 (22.2%)	0.82
1–2 L	32 (28%)	15 (33.3%)	
>2 L	53 (47%)	20 (44.5%)	

**Table 2 nutrients-16-01193-t002:** Comparison between IBD patients at risk of orthorexia and not at risk of orthorexia.

IBD Patients *N* = 113	Patients with the Risk of Orthorexia *N* = 26	Patients without the Risk of Orthorexia *N* = 87	*p*-Value
Age (years) (mean ± SD)	51 ± 17	49 ± 15	0.66
**Gender**			
Female	10 (38.4%)	42 (48.2%)	0.37
Male	16 (61.6%)	45 (51.8%)	
**Marital status**			0.78
Married	19 (73.1%)	54 (62.1%)	
Single/widowed/divorced	7 (26.9%)	33 (37.9%)	
BMI—Body Mass Index (mean ± SD)	26.7 ± 6.7	25.0 ± 4.5	0.12
**Educational qualifications**			
Primary-school degree	10 (38.4%)	38(43.7%)	0.88
Secondary-school degree	3 (11.6%)	13 (14.9%)	
High-school degree	10 (11.6%)	27 (31.0%)	
University degree	3 (14%)	9 (10.4%)	
**Employment**			
Employed	11 (42.3%)	41 (47.1%)	0.64
Unemployed	8 (30.7%)	27 (31.0%)	
Retired	6 (23.2%)	12 (13.8%)	
Student	1 (3.8%)	7 (8.1%)	
**Diagnosis**			
CD	11 (42.3%)	37 (42.5%)	0.98
UC	15 (57.7%)	50 (57.5%)	
**Disease duration**			
<5 years	11 (32.4%)	33 (37.9%)	0.75
5–10 years	4 (15.4%)	22 (25.3%)	
11–20 years	8 (30.7%)	22 (25.3%)	
>20 years	3 (11.5%)	10 (11.5%)	
**Therapy**			
Biologic agents	18 (69.2%)	67 (77.0%)	0.42
Conventional therapy	8 (30.8%)	20 (23.0%)	
**Partial Mayo score/HBI**			
Remission	11 (42.3%)	37 (42.5%)	0.47
Mild activity	9 (34.6%)	18 (20.8%)	
Moderate activity	5 (19.3%)	20 (22.9%)	
Severe activity	1 (3.8%)	12 (13.8%)	
**Surgery**			
Yes	1 (3.8%)	17 (19.5%)	
No	25 (96.2%)	70 (81.5%)	0.05
**Ostomy**			
No	0 (0%)	3 (3.4%)	
Yes	26 (100%)	84 (96.6%)	0.33
**Food intolerances**			
No	1 (3.8%)	5 (5.7%)	
Lactose intolerance/celiac disease	25 (96.2%)	82 (94.3%)	0.84
**Smoking habit**			
Yes	8 (30.7%)	30 (34.5%)	
No	18 (69.3%)	57 (65.5%)	0.72
**Daily water consumption**			
<1 L	6 (23.1%)	22 (25.3%)	0.80
1–2 L	6 (23.1%)	26 (29.9%)	
>2 L	14 (53.8%)	39 (44.8%)	

**Table 3 nutrients-16-01193-t003:** Comparison between IBD patients with risk of orthorexia and without risk of orthorexia (<30 years).

IBD Patients (<30 years) *N* = 14	Patients with the Risk of Orthorexia *N* = 11	Patients without the Risk of Orthorexia *N* = 3	*p*-Value
Age (years) (mean ± SD)	23 ± 3	23 ± 6	0.85
**Gender**			
Female	5 (45.5%)	1 (33.3%)	0.70
Male	6 (54.4%)	2 (66.7%)	
**Marital status**			
Married	0	1 (33.3%)	0.04
Single/widowed/divorced	11 (100%)	2 (66.7%)	
BMI—Body Mass Index (mean ± SD)	23 ± 3	21 ± 4	0.56
**Educational qualifications**			
Primary-school degree	1 (9.1%)	0	
Secondary-school degree	0 (%)	1 (33.3%)	0.21
High-school degree	8 (72.7%)	2 (66.7%)	
University degree	2 (18.2%)	0	
**Employment**			
Employed	3 (27.3%)	2 (66.7%)	
Unemployed	3 (27.3%)	0	
Retired	0 (%)	0	0.38
Student	5 (45.4%)	1 (33.3%)	
**Diagnosis**			
CD	9 (81.8%)	1 (33.3%)	0.09
UC	2 (18.2%)	2 (66.7%)	
**Disease duration**			
<5 years	6 (54.5%)	2 (66.7%)	
5–10 years	3 (27.3%)	1 (33.3%)	0.74
11–20 years	2 (18.2%)	0	
>20 years	0	0	
**Therapy**			
Biologic agents	10 (90.9%)	2 (66.7%)	0.28
Conventional therapy	1 (9.1%)	1 (33.3%)	
**Partial Mayo score/HBI**			
Remission	8 (72.7%)	2 (66.7%)	
Mild activity	2 (18.2%)	1 (33.3%)	0.81
Moderate activity	1 (9.1%)	0	
Severe activity	0	0	
**Surgery**			
Yes	3 (27.3%)	0	
No	8 (72.7%)	3 (100%)	0.76
**Ostomy**			
No	11 (100%)	3 (100%)	1
Yes	0	0	
**Food intolerances**			
No	8 (72.7%)	3 (100%)	0.30
Lactose intolerance/celiac disease	3 (27.3%)	0	
**Smoking habit**			
Yes	5 (45.4%)	1 (33.3%)	
No	6 (54.5%)	2 (66.7%)	0.82
**Daily water consumption**			
<1 L	2 (18.2%)	1 (33.3%)	0.72
1–2 L	8 (72.7%)	2 (66.7%)	
>2 L	1 (9%)	0	

**Table 4 nutrients-16-01193-t004:** Comparison between IBD patients with risk of orthorexia and without risk of orthorexia (≥30 years).

IBD Patients (≥30 years) *N* = 99	Patients with the Risk of Orthorexia *N* = 76	Patients without the Risk of Orthorexia *N* = 23	*p*-Value
Age (years) (mean ± SD)	54 ± 13	55 ± 15	0.65
**Gender**			
Female	37 (48.7%)	9 (39%)	
Male	39 (51.3%)	14 (61%)	0.42
**Marital status**			
Married	54 (71%)	18 (78.3%)	0.88
Single/widowed/divorced	22 (29%)	5 (21.7%)	
BMI—Body Mass Index (mean ± SD)	25 ± 5	27 ± 7	0.08
**Educational qualifications**			
Primary-school degree	8 (10.5%)	3 (13%)	
Secondary-school degree	27 (35.5%)	9 (39.1%)	
High-school degree	30 (39.5%)	8 (34.8%)	0.96
University degree	11 (14.5%)	3 (13%)	
**Employment**			
Employed	38 (50%)	9 (39.1%)	
Unemployed	24 (31.6%)	8 (34.8%)	
Retired	12 (15.8%)	6 (26%)	0.54
Student	2 (2.6%)	0	
**Diagnosis**			
CD	28 (36.8%)	10 (43.5%)	0.56
UC	48 (63.2%)	13 (56.5%)	
**Disease duration**			
<5 years	27 (35.5%)	9 (39.1%)	
5–10 years	19 (25%)	3 (13%)	0.73
11–20 years	20 (26.3%)	8 (34.8%)	
>20 years	10 (13.2%)	3 (13%)	
**Therapy**			
Biologic agents	57 (75%)	16 (69.6%)	0.60
Conventional therapy	19 (25%)	7 (30.4%)	
**Partial Mayo score/HBI**			
Remission	38 (50%)	16 (69.6%)	
Mild activity	34 (44.7%)	7 (30.4%)	0.63
Moderate activity	3 (3.9%)	*0*	
Severe activity	1 (1.3%)	*0*	
**Surgery**			
Yes	17 (22.4%)	1 (4.3%)	
No	59 (77.6%)	22 (95.7%)	0.31
**Ostomy**			
No	73 (96%)	23 (100%)	
Yes	3 (4%)	0	0.87
**Food intolerances**			
No	74 (97.4%)	22 (95.7%)	0.57
Lactose intolerance/celiac disease	2 (2.6%)	1 (4.3)	
**Smoking habit**			
Yes	25 (33%)	7 (30.4%)	0.89
No	51 (67%)	16 (69.6%)	
**Daily water consumption**			
<1 L	20 (26%)	5 (21.7%)	0.87
1–2 L	56 (74%)	18 (78.3%)	
>2 L	0	0	

## Data Availability

The data presented in this study are available on request from the corresponding author due to privacy reasons.

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
