# Peer review of "Food Beliefs and the Risk of Orthorexia in Patients with Inflammatory Bowel Disease"

_nutrients, 2024, doi:10.3390/nu16081193_

Round 1

Reviewer 1 Report

Comments and Suggestions for Authors

This article refers to the occurrence of orthorexia nervosa in patients with inflammatory bowel diseases.

·       The ORTO-15 questionnaire is a tool used to examine the risk/tendency to develop orthorexia. There are no official criteria for diagnosing orthorexia, so you cannot diagnose it only based on the ORTO-15 questionnaire. In abstract you write “Many patients with IBD develop orthorexia nervosa that impacts the patient's psychological and social sphere, exposes them to a high risk of nutritional deficiencies, and affects quality of life.” - You can't draw these conclusions just based on your research.

The authors themselves note that the study group is small. Therefore, the results cannot be generalized – “In conclusion, our study showed that most patients with IBD (77%) have a risk of developing orthorexia and that this is associated with a lower BMI and a history of previous IBD surgery” - In this part, the authors correctly note that based on their study, it is possible to assess the "risk of developing orthorexia", while this fragment shows that a total of  77% of patients with IBD have an increased risk, and based on this study on a non-representative group, it can only be concluded that the risk is probably increased in patients with IBD and add information that in the case of this research it was 77% of the study group.

·       The language needs corrections,

·       the authors do not pay attention to details - for example, some of the vocabulary in the tables is in Italian,

·       “Results” should be divided into subsections to increase clarity.

·       It would also be interesting if the authors present individual results from the ORTO-15 questionnaire - for example, the % distribution of often/always responses in the group of patients with and without risk of orthorexia.

·       the figures are cut off, in figure 3 individual elements are moved, making reading impossible,

·       in paragraph 39 the abbreviation "IOBD" without explanation,

·       in paragraphs 39-43 it is not explained to which group of patients the recommendations are addressed - for patients with active disease, in remission or perhaps for prevention?

·       Paragraphs 44-51 contain a sentence that too long. It can be assumed that the author has in mind factors that prevent the occurrence of IBD - to increase clarity, data should be listed in bullet points or presented in table form.

·       incomprehensible sentence "Disordered eating and eating disorders in IBD..." - paragraphs 56,

·       further too long sentences - paragraphs 59-64,

·       in the sentence "Among the most avoided 64 foods there are foods containing preservatives, additives, dyes, food flavorings, excess fat, sugar, salt, or genetically modified foods [12,13]." in paragraphs 64-66, the word "food" is used 4 times,

·       in paragraph 70-71 "The average prevalence rate of orthorexia is 6.9% for the general population and 35-57.8% for high-risk groups." - there is no explanation who belongs to the "high risk group",

·       paragraphs 113-124 present questions from the ORTO-15 questionnaire, which are also presented in Figure 3 - unnecessary repetition,

·       in paragraphs 172 and 173 "On the basis of the results of the Donini questionnaire with the cut-off of 40, we found that patients with IBD had a 77% risk of orthorexia, significantly higher than the 47% observed in the control group (p<0.001). - it cannot be stated that the risk of developing orthorexia is 77% and 47%, respectively, but only the percentages of people with a tendency to develop orthorexia in the studied groups.

·       Paragraphs 193 – “We have also documented how in the context of patients with IBD the presence of orthorexia is associated with lower BMI….” - information that this comparison concerns the control group should be added, because when comparing people with IBD with and without a tendency to orthorexia, this difference was not statistically significant.

Comments on the Quality of English Language

The article cannot be accepted in this form.

Reviewer 2 Report

Comments and Suggestions for Authors

The study presents a hypothesis about nervous orthorexia associated with patients with IBD. The study does not present ethical issues of the study for carrying out the study, even though they are questionnaires. It is not clear how the controls were selected. The tables have grammatical and interpretation errors. Furthermore, it is not possible to conclude that patients with IBD have this behavior, considering that other studies have shown that patients with IBD consume even more processed and ultra-processed products than controls. The discussion without pointing out a deeper approach. Long excerpts without reference are a description of the results without said discussion.

Round 2

Reviewer 2 Report

Comments and Suggestions for Authors

The authors have improved the text by providing justifications for several points. However, some aspects need clarification.

·      There is significant variability in the age range between patients with IBD and those in the control group. This variability in such studies may introduce bias, especially considering that the instrument used is a survey. Therefore, I suggest stratifying the tables presented by at least 2 or 3 age groups to account for this variability.

·       Additionally, it would be beneficial to clarify to the reader how the questionnaire is administered and how scores ranging from 1 to 4 are assigned to the questions.

·       Another issue that should be acknowledged as a limitation of the study is the potential for biased responses due to incorrect administration of the questionnaires. Therefore, it is essential to explain how the questionnaires were administered and who conducted the administration.

·       The entire text needs to be reviewed, as there are instances where it is evident that text from another document was added, resulting in variations in fonts and sizes. It is important to ensure consistency in font and size throughout the document.
